# Rapid epigenetic adaptation to uncontrolled heterochromatin spreading

Jiyong Wang, Bharat D Reddy, Songtao Jia*

Department of Biological Sciences, Columbia University, New York, United States

**Abstract** Heterochromatin, a highly compact chromatin state characterized by histone H3K9 methylation and HP1 protein binding, silences the underlying DNA and influences the expression of neighboring genes. However, the mechanisms that regulate heterochromatin spreading are not well understood. In this study, we show that the conserved Mst2 histone acetyltransferase complex in fission yeast regulates histone turnover at heterochromatin regions to control heterochromatin spreading and prevents ectopic heterochromatin assembly. The combined loss of Mst2 and the JmjC domain protein Epe1 results in uncontrolled heterochromatin spreading and massive ectopic heterochromatin, leading to severe growth defects due to the inactivation of essential genes. Interestingly, these cells quickly recover by accumulating heterochromatin at genes essential for heterochromatin assembly, leading to their reduced expression to restrain heterochromatin spreading. Our studies discover redundant pathways that control heterochromatin spreading and prevent ectopic heterochromatin assembly and reveal a fast epigenetic adaptation response to changes in heterochromatin landscape.

*For correspondence: jia@
biology.columbia.edu

**Competing interests:** The
authors declare that no
competing interests exist.

**Reviewing editor**: Ali Shilatifard,
Northwestern University
Feinberg School of Medicine,
United States

## Introduction

Eukaryotic genomic DNA is folded with histones and non-histone proteins in the form of chromatin, which regulates every aspect of DNA metabolism, including transcription, replication, recombination, and DNA damage repair. Chromatin is classified into euchromatin, which is gene rich and actively transcribed, and heterochromatin, which is gene poor and highly compacted (*Grewal and Jia, 2007*). Heterochromatin preferentially forms at repetitive DNA elements in order to limit transcription and recombination at these regions to maintain genome integrity. It also forms at developmentally regulated genes to regulate their expression in response to developmental cues and external stimuli. Heterochromatin tends to spread into neighboring regions, leading to the inactivation of genes in a sequence-independent manner (*Talbert and Henikoff, 2006*; *Wang et al., 2014*). Therefore, the sites of heterochromatin formation and extent of heterochromatin spreading need to be tightly controlled to prevent improper gene silencing, and misregulation of heterochromatin assembly has been linked to many human diseases, especially various types of cancers (*Geutjes et al., 2012*).

Heterochromatic regions generally have distinct chromatin signatures such as histones that are hypoacetylated and methylated at histone H3 lysine 9 (H3K9me), and the enrichment of HP1 family proteins (*Rea et al., 2000*; *Bannister et al., 2001*; *Lachner et al., 2001*; *Nakayama et al., 2001*). Formation of heterochromatin requires the concerted actions of a diverse group of histone-modifying proteins, such as H3K9 methyltransferases and histone deacetylases (HDACs), and is divided into three distinct steps: establishment, spreading, and maintenance (*Grewal and Moazed, 2003*; *Rusche et al., 2003*). Heterochromatin is established at nucleation centers through the targeting of histone-modifying activities by transcription factors or non-coding RNAs (*Cohen and Jia, 2014*). Subsequently, heterochromatin spreads into neighboring regions, mostly via a network of interactions among chromatin proteins, resulting in the formation of large heterochromatin domains independent of the underlying DNA sequences (*Talbert and Henikoff, 2006*; *Cohen and Jia, 2014*). Once these

**eLife digest** The DNA in the nucleus of a cell is wrapped around histone proteins to form a compact structure known as chromatin. Chromatin's structure can control how the genes in DNA are expressed. Loosely packed chromatin contains active genes, whereas densely packed chromatin (also called 'heterochromatin') contains silenced genes that are not expressed. The assembly of DNA into heterochromatin needs to be carefully controlled. Otherwise, the DNA next to heterochromatin regions can become densely packed as well (via a process called 'heterochromatin spreading'), and the genes within this DNA are incorrectly silenced. Incorrect gene silencing is often associated with diseases such as cancer.

Cells add chemical groups onto the histone proteins to influence how chromatin is compacted. Densely packed chromatin contains histones with many methyl groups but few acetyl groups. A protein called Epe1, which potentially removes methyl groups, helps to prevent heterochromatin spreading in yeast cells. Wang et al. found that an enzyme called Mst2, which adds acetyl groups onto histones, also limits heterochromatin spreading and prevents extra heterochromatin from assembling at undesirable locations.

Wang et al. then generated yeast cells that lacked both Epe1 and Mst2. At first, these cells were sickly and unable to grow, because several essential genes were incorrectly silenced due to rampant heterochromatin spreading. However, the cells quickly overcame this growth defect by gaining an additional mutation. Normally mutations occur through changes in DNA sequences. However, Wang et al. found that the cells acquired this mutation by packing a gene required for heterochromatin assembly into heterochromatin. This in turn stopped more chromatin from becoming packed too densely. Changes to chromatin can also be passed on to the yeast's offspring, and such a change could help the offspring to better cope with changes in heterochromatin levels. Future work could test how often inheritable changes to chromatin modification help organisms adapt to environmental stresses, or if similar changes allow cancer cells to become tolerant to anticancer drugs.

domains are formed, they can maintain themselves also through interactions among chromatin proteins even in the absence of the initiation signal (*Moazed, 2011*; *Ragunathan et al., 2014*).

The formation of heterochromatin has been extensively studied in fission yeast, which uses highly conserved histone-modifying enzymes and chromatin proteins for heterochromatin assembly, such as the SUV39 family histone H3K9 methyltransferase Clr4, the HP1 homologue Swi6, and HDACs Sir2 and Clr3 (*Grewal and Jia, 2007*). There are four types of heterochromatin identified in fission yeast: constitutive heterochromatin at repeat regions such as centromeres, telomeres, and the silent mating-type region (*Grewal and Jia, 2007*); facultative heterochromatin islands at a subset of meiotic genes (*Hiriart et al., 2012*; *Zofall et al., 2012*; *Tashiro et al., 2013*; *Egan et al., 2014*); HOODs (heterochromatin domains) at sexual differentiation genes and retrotransposons in response to the misregulation of the exosome (*Yamanaka et al., 2013*); and transient heterochromatin at convergent genes (*Gullerova and Proudfoot, 2008*). These locations use distinct pathways to recruit histone-modifying activities to form heterochromatin.

The establishment of constitutive heterochromatin at repetitive DNA elements requires the RNA interference (RNAi) pathway, a phenomenon also vastly conserved in eukaryotes (*Moazed, 2009*; *Lejeune and Allshire, 2011*; *Castel and Martienssen, 2013*). The DNA repeats are transcribed and the transcripts are processed by the RNAi machinery into small interfering RNAs (siRNAs), which target the Clr4 complex (CLRC, consisting of Clr4, Cul4, Rik1, Raf1, and Raf2) to repeat regions to initiate H3K9me. In addition, DNA binding factors, such as telomeric shelterin and stress-activated ATF/CREB family proteins Atf1/Pcr1, also directly recruit histone-modifying activities to establish constitutive heterochromatin at telomeres and the silent mating-type region, respectively (*Jia et al., 2004*; *Kim et al., 2004*; *Kanoh et al., 2005*; *Tadeo et al., 2013*). The formation of facultative heterochromatin islands at meiotic genes requires RNA binding protein Mmi1, Zinc finger protein Red1, and the exosome (*Hiriart et al., 2012*; *Zofall et al., 2012*; *Tashiro et al., 2013*; *Egan et al., 2014*). Mmi1 binds to RNA transcripts containing DSR (determinant of selective removal) sequences and recruits the RNA-induced transcriptional silencing (RITS) complex and the Red1-Mtl1 complex, which directly interacts with Clr4 complex, to initiate H3K9me at meiotic genes (*Harigaya et al., 2006*;

*Zofall et al., 2012*; *Lee et al., 2013*; *Egan et al., 2014*). HOODs are formed at sexual differentiation genes and retrotransposons in response to exosome malfunction or changes in environmental conditions and requires RNAi, polyA polymerase Pla1, and PolyA binding protein Pab2 (*Lee et al., 2013*; *Yamanaka et al., 2013*). Convergent genes generate overlapping transcripts during the G1 phase of the cell cycle, which induce the formation of transient heterochromatin through the RNAi pathway (*Gullerova and Proudfoot, 2008*).

The spreading of heterochromatin requires Swi6 and the chromodomain of Clr4, both of which bind to H3K9me and position Clr4 to methylate neighboring nucleosomes (*Hall et al., 2002*; *Zhang et al., 2008*; *Al-Sady et al., 2013*). The reiteration of H3K9 methylation and recruitment of Clr4 by H3K9me, either directly through the chromodomain or indirectly through Swi6, results in the 'inch-worm'-like spreading of heterochromatin from nucleation centers into large chromosome domains (*Talbert and Henikoff, 2006*; *Wang et al., 2014*). Some heterochromatin regions are flanked by DNA sequences termed boundary elements, which block heterochromatin spreading (*Wang et al., 2014*). In other cases, heterochromatin borders are determined by the local balance of heterochromatin and euchromatin proteins, which tends to differ between cells. Therefore, precise regulation of heterochromatin spreading is essential to maintain stable gene expression profiles.

One of the best-known negative regulators of heterochromatin spreading is Epe1 as *epe1Δ* results in heterochromatin spreading beyond its normal boundaries as well as ectopic heterochromatin formation (*Zofall and Grewal, 2006*; *Trewick et al., 2007*; *Zofall et al., 2012*; *Ragunathan et al., 2014*). Loss of Epe1 also bypasses RNAi for pericentric heterochromatin assembly by strengthening heterochromatin spreading (*Trewick et al., 2007*). Epe1 contains a JmjC domain, which is frequently associated with histone demethylase activity. Although no demethylase activity has been detected for Epe1 (*Tsukada et al., 2006*), genetic evidence is consistent with Epe1 being a H3K9 demethylase and conserved catalytic residues are essential for Epe1 function (*Trewick et al., 2007*; *Ragunathan et al., 2014*).

The Mst2 complex is similar in composition to budding yeast NuA3 and mammalian HBO1/MOZ/MORF complexes (*Wang et al., 2012*). It is a highly specific histone H3K14 acetyltransferase that cooperates with Gcn5 to regulate global H3K14 acetylation levels (*Nugent et al., 2010*; *Wang et al., 2012*). The formation of heterochromatin is negatively correlated with H3K14 acetylation (*Sugiyama et al., 2007*; *Motamedi et al., 2008*), and *mst2Δ* bypasses the requirement of the RNAi pathway for pericentric heterochromatin assembly through modulating H3K14ac levels at heterochromatin (*Reddy et al., 2011*). Moreover, *mst2Δ* strengthens silencing at telomeres (*Gomez et al., 2005*). These results suggest that Mst2 complex functions to antagonize heterochromatic silencing, although the mechanism by which it affects heterochromatin assembly is unknown. The ability to bypass RNAi requires ablating the enzymatic activity of the Mst2 complex (*Reddy et al., 2011*). It was proposed that Mst2-mediated H3K14 acetylation regulates histone turnover at heterochromatin regions and the loss of such activity preserves parental histone modifications to promote heterochromatin maintenance (*Reddy et al., 2011*), although the ability of Mst2 to regulate histone turnover has not been directly tested.

In this study, we show that Mst2 regulates histone turnover at heterochromatin regions and that loss of Mst2 results in heterochromatin spreading at telomeres and heterochromatin islands where boundaries are absent. We also found that *mst2Δ epe1Δ* cells are initially very sick due to heterochromatin spreading-mediated inactivation of essential genes, suggesting that Mst2 and Epe1 function redundantly in regulating heterochromatin spreading. Interestingly, these cells quickly recover by forming ectopic heterochromatin at the *clr4+* locus to mitigate the negative effects of heterochromatin. Disrupting heterochromatin assembly at the *clr4+* locus results in ectopic heterochromatin formation at the *rik1+* locus, which encodes another subunit of the Clr4 complex required for H3K9me. These results demonstrate that promiscuous heterochromatin assembly generates epigenetic mutations that provide fast adaptions to heterochromatin stress.

## Results

### Mst2 regulates histone turnover at heterochromatin

To directly examine the role of the Mst2 complex in regulating histone turnover, we generated a Flag-tagged version of histone H3 driven by the *urg1* promoter at the endogenous *urg1+* locus, which can be quickly induced by the addition of uracil into the growth medium at levels far below the endogenous histone H3 (*Watt et al., 2008*) (*Figure 1A*). To prevent replication-dependent histone incorporation, we blocked the cell cycle with hydroxyl urea (HU) before induction of H3-Flag

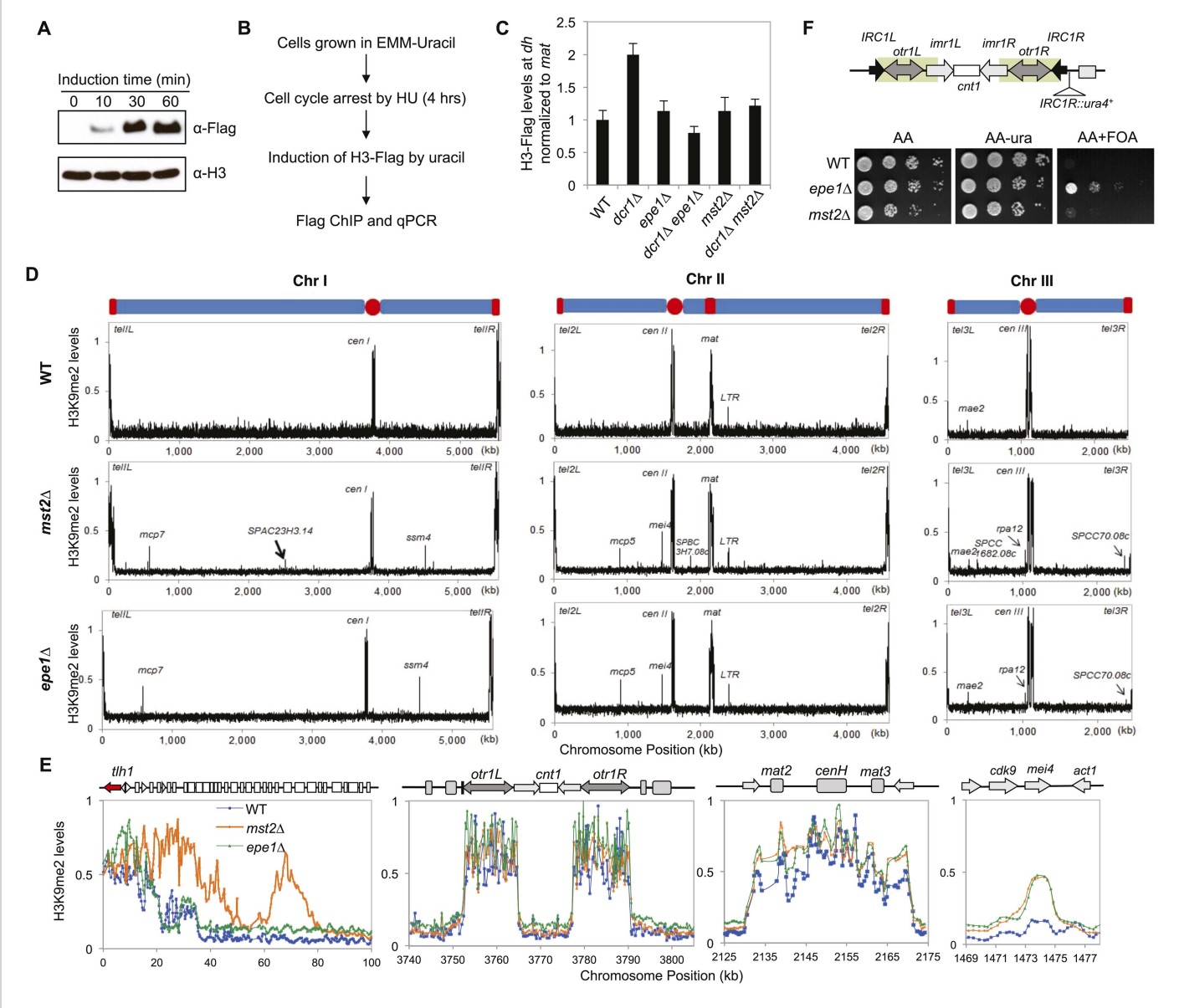

**Figure 1**. Mst2 counteracts heterochromatin assembly. (**A**) Western blot analysis of H3-Flag levels. Samples were taken at indicated times after the addition of uracil, and Western blot analyses were performed with Flag and H3 antibodies. (**B**) Schematic diagram of the histone turnover assay. (**C**) Enrichment of H3-Flag at pericentric *dh* sequence as an indicator of histone turnover rates. The values are normalized to a region within the silent mating-type locus with background histone turnover (*Aygun et al., 2013*). Error bars represent standard deviation of three experiments. (**D**) ChIP–chip analyses of H3K9me2 levels across the genome. (**E**) ChIP–chip data of H3K9me2 levels around the telomere IL, centromere I, the silent mating-type region, and the *mei4+* locus. (**F**) Mst2 is not required for boundary function at *IRC1R*. Serial dilution analysis were performed to measure the expression of *IRC1R::ura4+* reporter.

The following figure supplement is available for figure 1:

**Figure supplement 1**. ChIP–chip data of H3K9me2 levels around centromere II, centromere III, telomere 1R, telomere 2L, and telomere 2R.

expression (*Figure 1B*). We found that pericentric *dh* repeat was associated with lower amounts of H3-Flag in wild-type cells compared with RNAi mutant *dcr1Δ* (*Figure 1C*), suggesting that histone turnover rates increase when heterochromatin is compromised. In addition, the incorporation of H3-Flag was reduced in *epe1Δ dcr1Δ* cells, as observed previously (*Figure 1C*) (*Aygun et al., 2013*). In *mst2Δ dcr1Δ* cells, H3-Flag incorporation was reduced to wild-type levels (*Figure 1C*), suggesting that the Mst2 complex indeed regulates histone turnover at heterochromatin.

To further examine the role of the Mst2 complex in regulating heterochromatin assembly, we performed Chromatin Immunoprecipitation coupled with DNA microarray (ChIP–chip) analyses of H3K9me2 levels across the fission yeast genome. In wild-type cells, H3K9me2 was mainly present at centromeres, telomeres, and the silent mating-type region (*Figure 1D*). There were also a few heterochromatic islands with low levels of H3K9me2 (*Figure 1D*). Although less heterochromatic islands were identified compared to a recent study (*Zofall et al., 2012*), our results are consistent with that of an earlier one (*Cam et al., 2005*). The discrepancies might be caused by the use of batches of antibody with different sensitivity or different data processing methods. In *mst2Δ* cells, constitutive heterochromatin domains at centromeres and the silent mating-type region were in good agreement with wild-type cells, but telomeric heterochromatin showed significant spreading into chromosome arms (*Figure 1D,E* and *Figure 1—figure supplement 1*), consistent with previous findings that *mst2Δ* strengthens silencing at telomeres (*Gomez et al., 2005*). Interestingly, there are a number of additional small H3K9me2 peaks scattered across the genome, most of which are also present in *epe1Δ* cells (*Figure 1D,E*, and *Supplementary file 1*) (*Zofall et al., 2012*). Therefore, Mst2 also prevents ectopic heterochromatin assembly, similar to Epe1. We observed only minor heterochromatin spreading in telomeric regions in *epe1Δ* cells compared with a previous study (*Zofall et al., 2012*), which might be due to the presence of two epigenetically stable subpopulations of cells with different effects on heterochromatin assembly (*Trewick et al., 2007*).

The difference between pericentric regions, telomeres, and heterochromatin islands is the presence of well-defined boundary elements at pericentric regions that block heterochromatin spreading (*Wang et al., 2014*). We found that *mst2Δ* has no effect on boundary activity of an inverted repeat at the pericentric region, *IRC1R*, which requires Epe1 and the double bromodomain protein Bdf2 for function (*Figure 1F*) (*Wang et al., 2013*), suggesting that Mst2 regulates heterochromatin spreading only in the absence of boundaries.

## Misregulation of heterochromatin affects the fitness of *mst2Δ epe1Δ* cells

Since *mst2Δ* and *epe1Δ* have similar phenotypes in heterochromatin assembly and each bypasses the RNAi pathway for pericentric heterochromatin functions (*Trewick et al., 2007*; *Reddy et al., 2011*), we generated *mst2Δ epe1Δ* cells to examine their epistatic relationship. All freshly generated *mst2Δ epe1Δ* cells formed very small colonies, suggesting a strong negative genetic interaction between these two mutants (*Figure 2A* and *Figure 2—figure supplement 1*), consistent with high throughput epistasis mapping (*Roguev et al., 2008*; *Ryan et al., 2012*). Moreover, abolishing the enzymatic activity of Mst2 (*mst2-E274Q* or *nto1Δ*) or Epe1 (*epe1-H374A* and *epe1-Y307A*) resulted in similar sickness (*Figure 2—figure supplement 1*), suggesting that the enzymatic activities of Mst2 and Epe1 have redundant functions. Double mutant of *mst2Δ bdf2Δ* had no defects in growth (*Figure 2—figure supplement 1*), suggesting that the boundary activity of Epe1 is not involved in genetic interaction with Mst2.

Interestingly, cells from the small *mst2Δ epe1Δ* colonies grew comparably to wild-type cells (*Figure 2B,C*), suggesting that the accumulation of either a genetic or more intriguingly, an epigenetic suppressor leads to quick and persistent production of normally growing *mst2Δ epe1Δ* cells. We crossed independent clones of recovered *mst2Δ epe1Δ* cells (*mst2Δ epe1Δ**, which denotes the presence of a suppressor) and found that all resulting progenies grew normally from the beginning (*Figure 2D*). Moreover, when we crossed *mst2Δ epe1Δ** cells with wild-type cells, half of the resulting *mst2Δ epe1Δ* colonies were small and the other half were normal (*Figure 2E*), suggesting that changes associated with a single genomic locus was responsible for the recovery of these cells and also ruling out the possibility that the initial growth defects were the result of these cells going through meiosis.

To identify the suppressor, we performed two successive rounds of crosses to introduce *mst2Δ* and *epe1Δ* into the fission yeast deletion library and measured the initial growth of triple mutants before accumulation of a suppressor (*Figure 2F*). Interestingly, a number of heterochromatin assembly mutants allowed robust growth of *mst2Δ epe1Δ* cells. Such mutants included deletions of components of the CLRC histone H3K9 methyltransferase complex (*clr4Δ*, *raf1Δ*, and *raf2Δ*), an HP1 protein (*swi6Δ*), and histone deacetylases (*sir2Δ* and *clr3Δ*) (*Figure 2G* and *Supplementary file 2*). In contrast, none of the RNAi factors were identified in our screen. Individual crosses also confirmed that heterochromatin mutants conferred normal growth to *mst2Δ epe1Δ* cells (*Figure 2H* and *Figure 2—figure supplement 2*). Such data suggest that the effect of *mst2Δ epe1Δ* on cell growth is possibly the result of misregulation of heterochromatin.

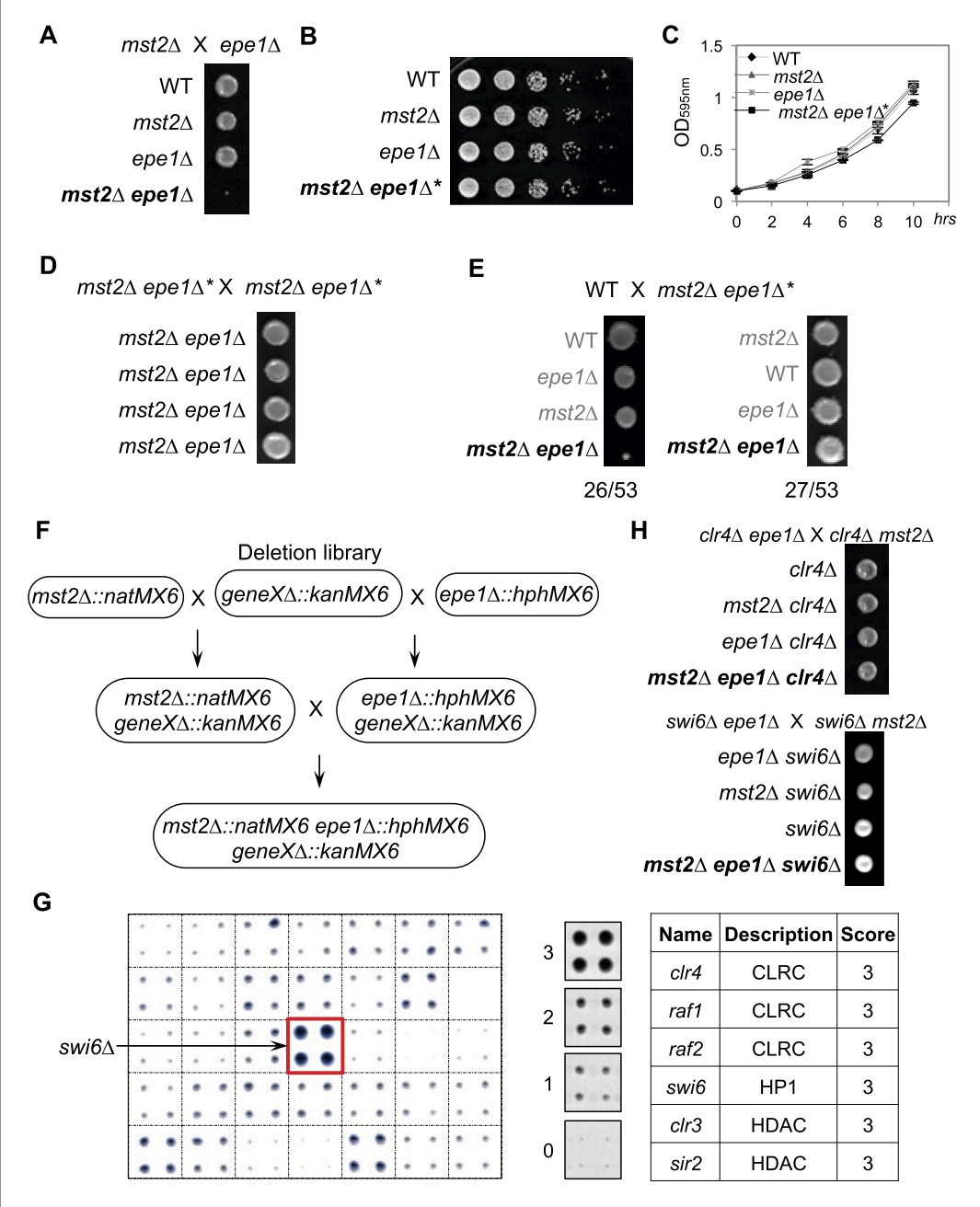

**Figure 2**. A suppressor mutation confers normal growth of *mst2Δ epe1Δ* cells. (**A**, **D**, **E**, **H**) Tetrad dissection analysis of the indicated genetic crosses. Pictures are examples of colonies derived from the same tetrad containing all individual genotypes, after one replication for a total of 6 days growth. (**B**) Serial dilution analysis of indicated strains. Cells were grown in rich medium overnight before dilution analyses were performed. (**C**) The growth curve of indicated strains. (**F**) Workflow to introduce *mst2Δ* and *epe1Δ* into the deletion library. (**G**) Left, a representative image of colony growth was shown. Middle, colonies were assigned scores between 0 and 3, as indicated. Right, list of identified heterochromatin mutants that confer fast growth.

The following figure supplements are available for figure 2:

**Figure supplement 1**. Tetrad dissection analysis of the indicated genetic crosses.

**Figure supplement 2**. Tetrad dissection analysis of indicated genetic crosses.

## Increased heterochromatin spreading is responsible for the initial growth defects of mst2Δ epe1Δ cells

To determine whether there are any global changes in heterochromatin organization, we decided to perform ChIP–chip analysis of H3K9me2 levels across the genome in mst2Δ epe1Δ cells. However, the quick generation of epigenetic suppressors in mst2Δ epe1Δ* cells prevented us from directly testing the reason for the initial growth defects. Because H3K9me functions upstream of Swi6 localization and the silencing function of H3K9me requires Swi6, we reasoned that examining H3K9me2 levels in mst2Δ epe1Δ swi6Δ cells might show the misregulation of heterochromatin assembly that resembles early stages mst2Δ epe1Δ cells. Indeed, in mst2Δ epe1Δ swi6Δ cells, the H3K9me2 domains at constitutive heterochromatin regions such as centromeres showed significant expansion, even when boundary elements are present (Figure 3A,B). In addition, many additional peaks of H3K9me2 were detected across the genome, at levels comparable to constitutive heterochromatin domains (Figure 3A,B, Figure 3—figure supplement 1, and Supplementary file 1). Most, but not all, of these additional sites correspond to previously described heterochromatin islands. Compared to wild-type, mst2Δ, or epe1Δ cells, these heterochromatic islands were also greatly expanded (Figure 3B and Figure 3—figure supplement 1). Given that Swi6 also contributes to heterochromatin spreading (Hall et al., 2002; Al-Sady et al., 2013), heterochromatin probably spreads over even longer distances when Swi6 is present. Interestingly, essential genes reside within or near some of the expanded H3K9me2 domains (Figure 3B and Figure 3—figure supplement 1), suggesting that misregulation of heterochromatin spreading inactivates essential genes and causes the initial sickness of mst2Δ epe1Δ cells.

If heterochromatin spreading is the cause of the initial growth defects of mst2Δ epe1Δ cells, we expect that mutations blocking heterochromatin spreading will abolish such an effect. The spreading of heterochromatin requires Swi6 as well as the chromodomain of Clr4, which binds to pre-existing H3K9me to allow the modification of adjacent nucleosomes (Hall et al., 2002; Zhang et al., 2008; Al-Sady et al., 2013). Our data that mst2Δ epe1Δ swi6Δ allows normal growth (Figure 2H) is consistent with such a hypothesis. However, Swi6 is required for heterochromatin spreading as well as heterochromatin-mediated silencing, making it difficult to definitively assess the contribution of heterochromatin spreading in this process. We therefore tested a mutation within the Clr4 chromodomain, W31G, which affects the binding of Clr4 to H3K9me to block heterochromatin spreading (Zhang et al., 2008). Indeed, mst2Δ epe1Δ clr4-W31G cells showed no initial growth defects (Figure 3C) and heterochromatin expansion is prevented as indicated by ChIP analysis of H3K9me2 levels outside of centromere I boundary and at the mei4+ locus (Figure 3D).

## Ectopic heterochromatin at the clr4+ locus promotes adaption of mst2Δ epe1Δ cells

ChIP–chip analysis also showed that the patterns and levels of H3K9me2 in mst2Δ epe1Δ* cells were more similar to those in wild-type cells with high levels of H3K9me2 at constitutive heterochromatic regions and low levels of H3K9me2 at heterochromatic islands, and much less heterochromatin spreading compared with mst2Δ epe1Δ swi6Δ cells (Figure 3A,B). Most significantly, H3K9me2 was enriched at a 5 kilobase region covering clr4+ and an adjacent gene meu6+, at levels comparable to constitutive heterochromatin regions in independent clones tested (Figures 3A,B, and 4A, and data not shown). Consistent with the fact that H3K9me is associated with gene silencing, both clr4+ mRNA and Clr4 protein levels were reduced in mst2Δ epe1Δ* cells (Figure 4B,C).

To test if the epigenetically silenced clr4+ locus is the suppressor of mst2Δ epe1Δ* cells, we crossed mst2Δ epe1Δ* (containing a methylated clr4+ locus) with Flag-clr4+ cells (containing an unmethylated Flag-clr4+ locus). We found that the resulting mst2Δ epe1Δ clr4+ daughters (inheriting the methylated clr4+ locus) grew normally, whereas the mst2Δ epe1Δ Flag-clr4+ daughters (inheriting the unmethylated Flag-clr4+ locus) were sick (Figure 4D). Moreover, ChIP analysis showed that the mst2Δ epe1Δ clr4+ progeny also inherited H3K9me2 associated with this locus (Figure 4E), and clr4+ mRNA levels were low (Figure 4F). Therefore, the inheritance of a silenced clr4+ locus allows cells to avoid the negative effects on cell growth imposed by mst2Δ epe1Δ. In contrast, wild-type cells inheriting the methylated clr4+ locus lost H3K9me2, clr4+ mRNA levels were significantly restored, and these cells exhibited no defects in the silencing of a pericentric otr::ura4+ reporter gene (Figure 4E,F, G). These results suggest that the silencing of clr4+ is epigenetic and not due to changes in DNA sequence. They also suggest that the continued absence of Mst2 and Epe1 is required to maintain H3K9me2 at the clr4+ locus.

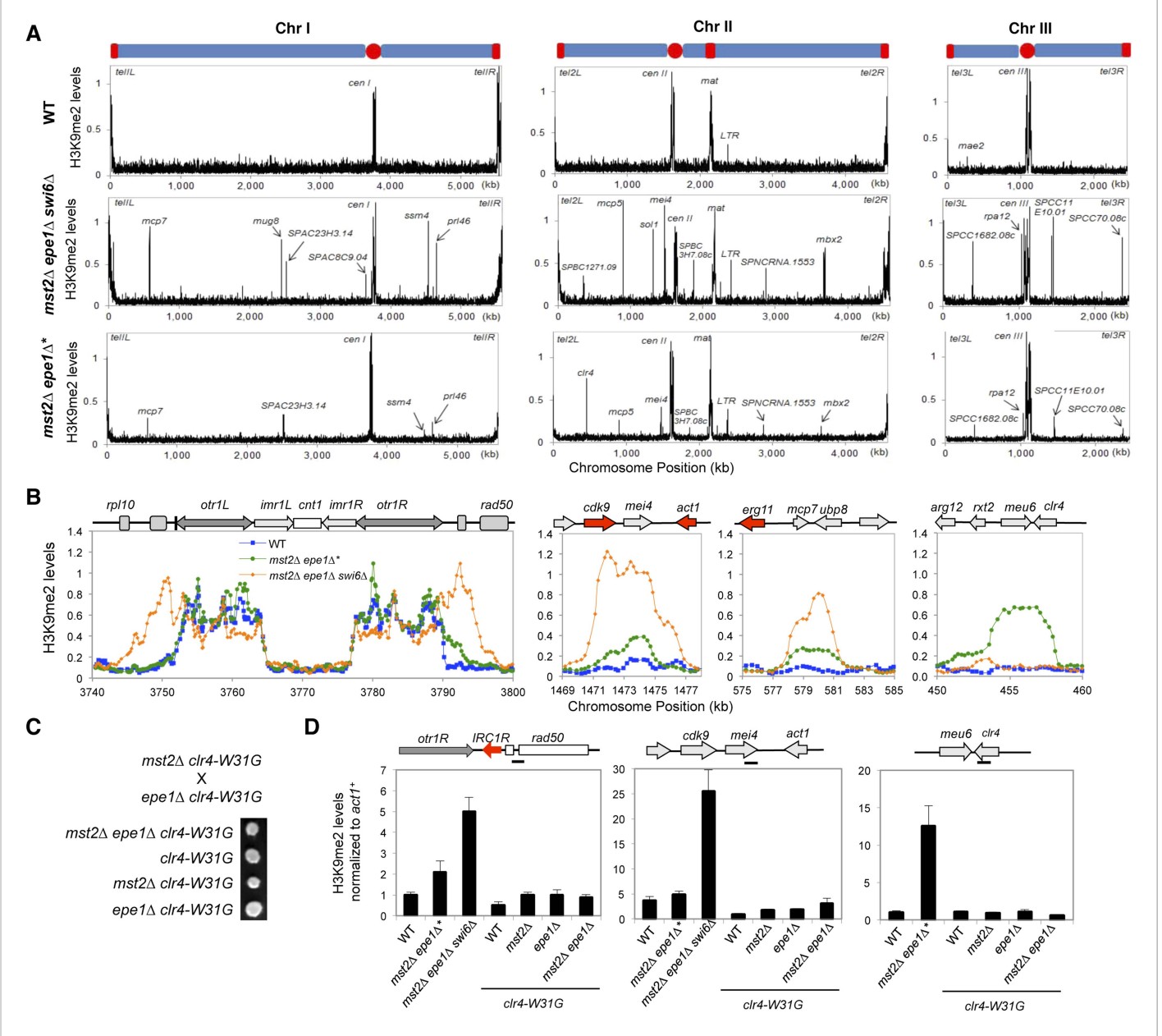

**Figure 3**. Increased heterochromatin spreading in *mst2Δ epe1Δ* cells leads to growth defects. (**A**) ChIP–chip analyses of H3K9me2 levels across the genome. (**B**) ChIP–chip data of H3K9me2 levels at centromere I, *mei4⁺*, *mcp7⁺*, and *clr4⁺* locus. (**C**) Tetrad dissection analysis of the indicated genetic cross. (**D**) ChIP-qPCR analysis of H3K9me2 levels at indicated locations, normalized against *act1⁺*. Error bars represent standard deviation of three experiments.

The following figure supplement is available for figure 3:

**Figure supplement 1**. ChIP–chip data of H3K9me2 levels at heterochromatin islands.

## Sequences 3′ to *clr4⁺* is required for heterochromatin assembly at the *clr4⁺* locus in *mst2Δ epe1Δ* cells

We then examined whether any of the known heterochromatin assembly pathways are required for heterochromatin assembly at the *clr4⁺* locus in *mst2Δ epe1Δ* cells. We found that *mst2Δ epe1Δ dcr1Δ* cells also quickly recovered and H3K9me2 levels were similar at the *clr4⁺* locus in *mst2Δ epe1Δ\** and *mst2Δ epe1Δ dcr1Δ\** cells, suggesting that RNAi is not required for heterochromatin assembly at *clr4⁺*, even though *clr4⁺* is in a convergent orientation with *meu6⁺* (**Figure 5—figure supplement 1**). In

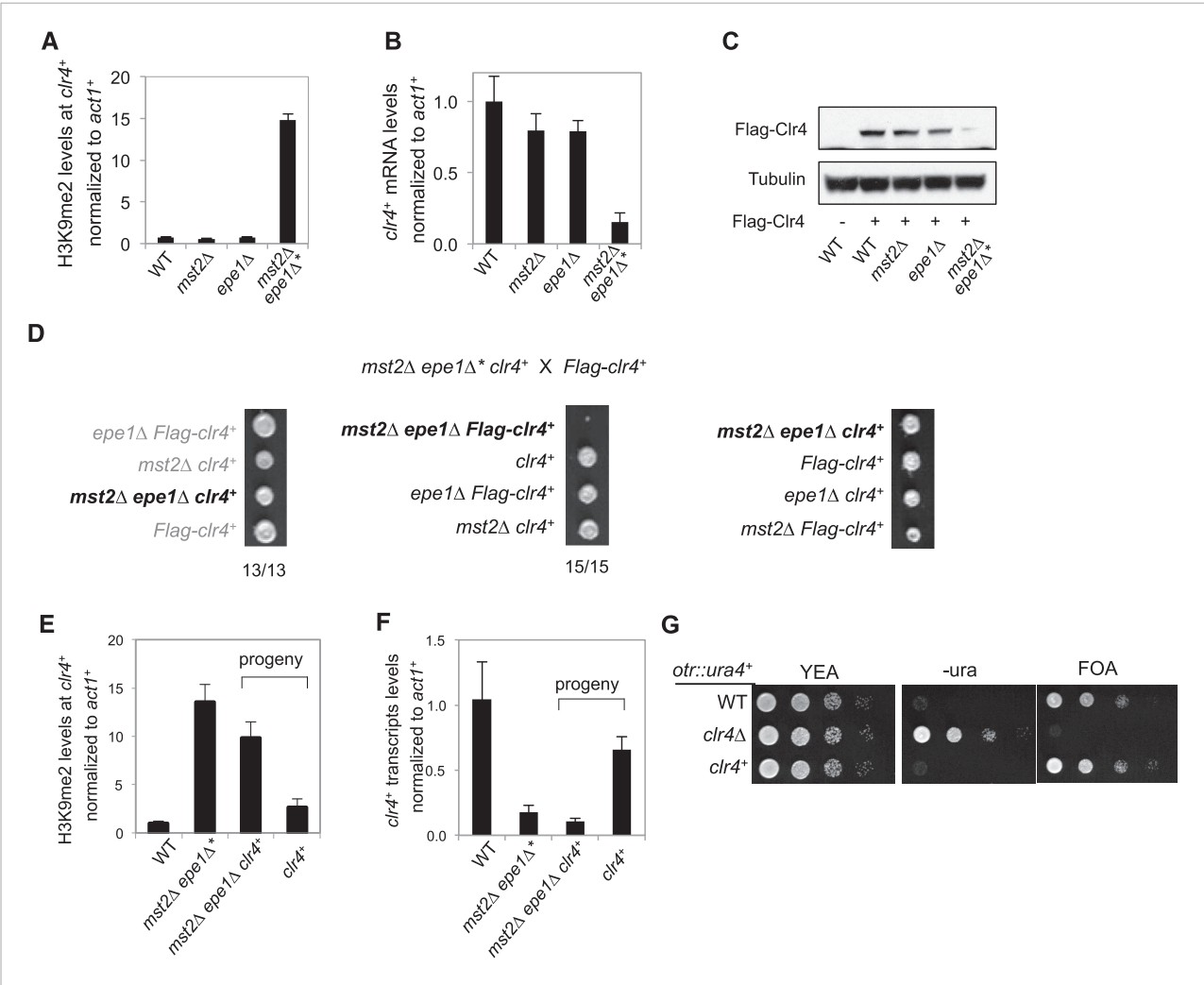

**Figure 4**. Inheritance of ectopic heterochromatin at the *meu6-clr4* locus. (**A**, **E**) ChIP-qPCR analysis of H3K9me2 levels at the *clr4+* coding region, normalized against *act1+*. Error bars represent standard deviation of three experiments. (**B**, **F**) qRT-PCR analysis of *clr4+* mRNA levels, normalized against *act1+*. Error bars represent standard deviation of three experiments. (**C**) Western blot analyses of Flag-Clr4 and Tubulin protein levels. (**D**) Tetrad dissection analysis of indicated genetic crosses. (**G**) Serial dilution analysis to measure the expression of *otr::ura4+* reporter.

addition, H3K9me2 levels persisted in *mst2Δ epe1Δ mmi1Δ\** and *mst2Δ epe1Δ pab2Δ\** cells (*Figure 5—figure supplement 1*), suggesting that heterochromatin assembly is not through Mmi1-mediated facultative heterochromatin assembly pathway or Pab2-mediated assembly of HOOD, even though *meu6+* is a meiotic gene. Therefore, heterochromatin assembly at *clr4+* differs from known heterochromatin assembly pathways. Due to the severe growth defects associated with *red1Δ* or *rrp6Δ*, we were unable to generate triple mutant strains with *mst2Δ epe1Δ* and whether these factors are involved in H3K9me2 at the *clr4+* locus in *mst2Δ epe1Δ* cells is unknown.

The domain of H3K9me2 in *mst2Δ epe1Δ\** cells includes *clr4+* and *meu6+*, with its center within *meu6+* coding region. Interestingly, RNA sequencing analysis showed that *meu6+* is not expressed in vegetative growing cells and *clr4+* transcript runs through the entire *meu6+* gene (*Figure 5A*) (*Wang et al., 2015*). We therefore replaced the entire *meu6+* open reading frame with a *kanMX6* cassette, in the same transcription orientation as *meu6+* (*Figure 5B*). Given that *meu6+* is right next to *clr4+*, we first examined whether this manipulation affects Clr4 function. We found that *meu6Δ::kanMX6* has no silencing defects at *otr::ura4+*, and *clr4+* mRNA and Clr4 protein levels were similar to those of wild-type cells (*Figure 5—figure supplement 2*). ChIP analysis showed that H3K9me2 was abolished at the *clr4+* locus in *mst2Δ epe1Δ meu6Δ\** cells (*Figure 5C*), and *clr4+* mRNA and Clr4 protein levels

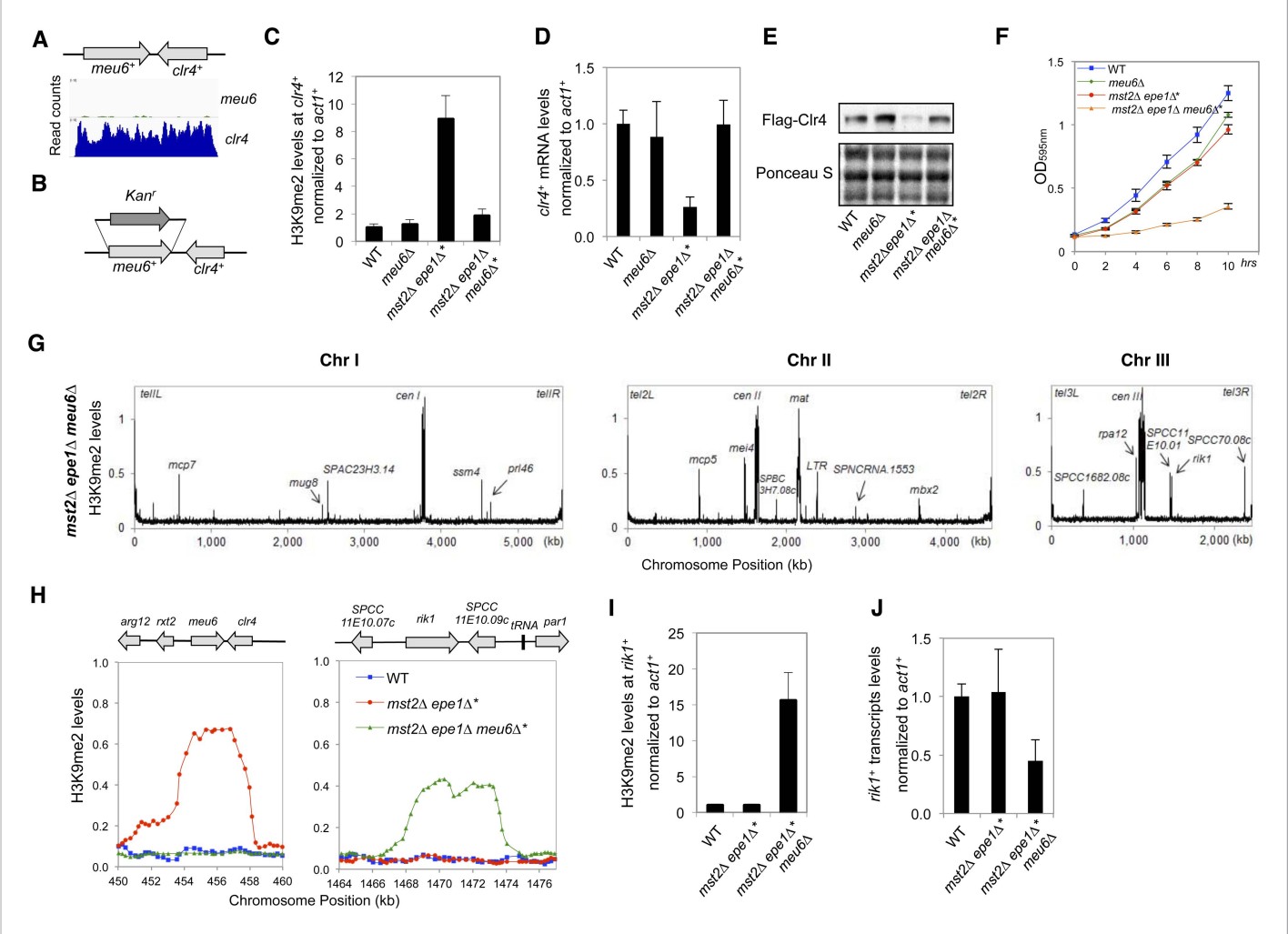

**Figure 5**. Blocking heterochromatin formation at the *clr4+* locus in *mst2Δ epe1Δ* cells results in ectopic heterochromatin assembly at the *rik1+* locus. (A) RNA sequencing data of the *meu6-clr4* region. (B) Schematic diagram of the *meu6Δ::kanMX6* construct. (C, I) ChIP-qPCR analysis of H3K9me2 levels at the *clr4+* or *rik1+* coding region, normalized against *act1+*. Error bars represent standard deviation of three experiments. (D, J) qRT-PCR analysis of *clr4+* or *rik1+* mRNA levels, normalized against *act1+*. Error bars represent standard deviation of three experiments. (E) Western blot analyses of Flag-Clr4 protein levels. (F) The growth curve of indicated strains. (G) ChIP–chip analyses of H3K9me2 levels across the genome in recovered *mst2Δ epe1Δ meu6Δ** cells. (H) ChIP–chip data of H3K9me2 levels at *clr4+* and *rik1+* loci.

The following figure supplements are available for figure 5:

**Figure supplement 1**. Dcr1, Mmi1 and Pab2 are not required for heterochromatin assembly at the *clr4+* locus in *mst2Δ epe1Δ** cells.

**Figure supplement 2**. *meu6Δ* has no effect on *clr4+* expression under normal conditions.

**Figure supplement 3**. Meu6-Flag abolished H3K9me in *mst2Δ epe1Δ* cells.

were similar to wild-type cells (*Figure 5D,E*). We also generated a *meu6-Flag::KanMX6* strain, which preserves the coding sequence of *meu6+*, and found that H3K9me2 levels at the *clr4+* locus were also abolished in *mst2Δ epe1Δ meu6-Flag::KanMX6* cells (*Figure 5—figure supplement 3*), suggesting that the *meu6+* coding sequence is not responsible for heterochromatin assembly at this locus, but rather the insertion of the *KanMX6* cassette disrupts a heterochromatin initiation signal. Given that the *clr4+* transcript has a long 3′-UTR and that both *meu6Δ::KanMX6* and *meu6-Flag::KanMX6* alter the 3′-UTR, heterochromatin assembly at the *clr4+* locus likely requires the intact 3′-UTR of *clr4+*.

## Disrupting ectopic heterochromatin assembly at the *clr4*⁺ locus results in ectopic heterochromatin assembly at the *rik1*⁺ locus

If the silenced *clr4*⁺ is the suppressor in *mst2Δ epe1Δ** cells, we expect that abolishing H3K9me2 at the *clr4*⁺ locus will result in the failure of *mst2Δ epe1Δ* cells to recover. Interestingly, *mst2Δ epe1Δ meu6Δ** cells were able to recover to some extent although they grew at a slower rate compared to wild-type or *mst2Δ epe1Δ** cells (**Figure 5F**). To understand how *mst2Δ epe1Δ meu6Δ** cells recover without silencing of *clr4*⁺, we performed ChIP–chip analysis of H3K9me2 levels in *mst2Δ epe1Δ meu6Δ** cells (**Figure 5G**). The distribution of H3K9me2 is on the whole similar to that of *mst2Δ epe1Δ** cells, with two major exceptions. First, H3K9me2 was indeed abolished from the entire *meu6*⁺ -*clr4*⁺ region (**Figure 5G,H**). Second, an additional peak of H3K9me2 appeared at a 7-kb region that includes two genes in convergent orientation: *rik1*⁺, which encodes a component of the Clr4 complex and is required for Clr4 function (**Nakayama et al., 2001**; **Hong et al., 2005**; **Horn et al., 2005**; **Jia et al., 2005**), and an uncharacterized gene *SPCC11E10.09c*⁺ (**Figure 5G,H,I**). As expected, *rik1*⁺ mRNA levels were significantly reduced in *mst2Δ epe1Δ meu6Δ** cells compared to wild-type or *mst2Δ epe1Δ** cells (**Figure 5J**). It is likely that the reduction of *rik1*⁺ expression allows *mst2Δ epe1Δ meu6Δ** cells to grow to some extent by decreasing global H3K9me and heterochromatin assembly. We suspect that *mst2Δ epe1Δ meu6Δ** cells recovered less well than *mst2Δ epe1Δ** cells because heterochromatin forms less efficiently at the *rik1*⁺ locus, which might explain why independent *mst2Δ epe1Δ** clones all preferentially silenced *clr4*⁺ (data not shown). Moreover, H3K9me2 was also enriched at the *rik1*⁺ locus in *mst2Δ epe1Δ meu6-Flag** cells (**Figure 5—figure supplement 3**), suggesting that the formation of heterochromatin at this locus is not random.

## Mst2 and Epe1 are required to counteract the high activity of Clr4

It is interesting to note that the enzymatic activity of Clr4 is much higher than that of its mammalian counterparts (**Rea et al., 2000**). Therefore, Mst2 and Epe1 are likely evolved to counteract such a hyperactive H3K9 methyltransferase. Clr4 has an arginine at residue 406, which corresponds to a histidine in its mammalian and *Drosophila* homologues (**Figure 6A**). Conversion of the histidine to arginine (H320R) makes mammalian SUV39H1 a hyperactive histone methyltransferase (**Rea et al., 2000**). We generated a R406H mutation in Clr4 and found that this mutation resulted in a drastic reduction of Clr4 enzymatic activity in vitro (**Figure 6B**). When introduced into the endogenous *clr4*⁺ locus, *R406H* moderately affected silencing of a pericentric *otr::ura4*⁺ reporter, indicating that high activity of Clr4 is required for heterochromatin assembly in wild-type cells (**Figure 6—figure supplement 1**). Interestingly, *mst2Δ epe1Δ clr4-R406H* had no initial growth defects (**Figure 6C**), no H3K9me2 at the *clr4*⁺ locus (**Figure 6D**), and *clr4*⁺ transcript levels were not affected (**Figure 6E**). Therefore, a less active Clr4 can also mitigate the effects of simultaneous loss of Mst2 and Epe1.

## Discussion

The formation of heterochromatin and its subsequent spreading result in silencing of large chromosomal domains in a sequence-independent manner. Therefore, the sites of heterochromatin assembly and the extent of heterochromatin spreading are generally precisely controlled to maintain stable gene expression patterns. In addition to the diverse pathways that accurately initiate heterochromatin assembly, anti-silencing activities also play essential roles in limiting heterochromatin spreading to shape the chromatin landscape.

Our results reveal a novel function of the Mst2 complex in regulating histone turnover at heterochromatin regions to counteract heterochromatin spreading. Loss of Mst2 bypasses the requirement of RNAi for pericentric heterochromatin assembly (**Reddy et al., 2011**), increases heterochromatin spreading and silencing at telomeres (**Figure 1**) (**Gomez et al., 2005**), and increases the efficiency of ectopic heterochromatin assembly (**Ragunathan et al., 2014**), which is phenotypically very similar to *epe1Δ*. Biochemically, the Mst2 complex is a highly specific histone H3K14 acetyltransferase and mutations resulting in the loss of its enzymatic activity, such as *mst2-E274Q* or *nto1Δ* (**Wang et al., 2012**), also resulted in slow growth when combined with *epe1Δ*, suggesting that its enzymatic activity is required for counteracting silencing. *H3K14* mutants have a direct effect on heterochromatin assembly independent of its acetylation state, making it difficult to directly address whether H3K14 is the only target of Mst2 in regulating heterochromatin spreading (**Mellone et al., 2003**; **Reddy et al., 2011**; **Alper et al., 2013**). Therefore, it remains possible that Mst2 modifies heterochromatin assembly factors to regulate histone turnover and counteract silencing.

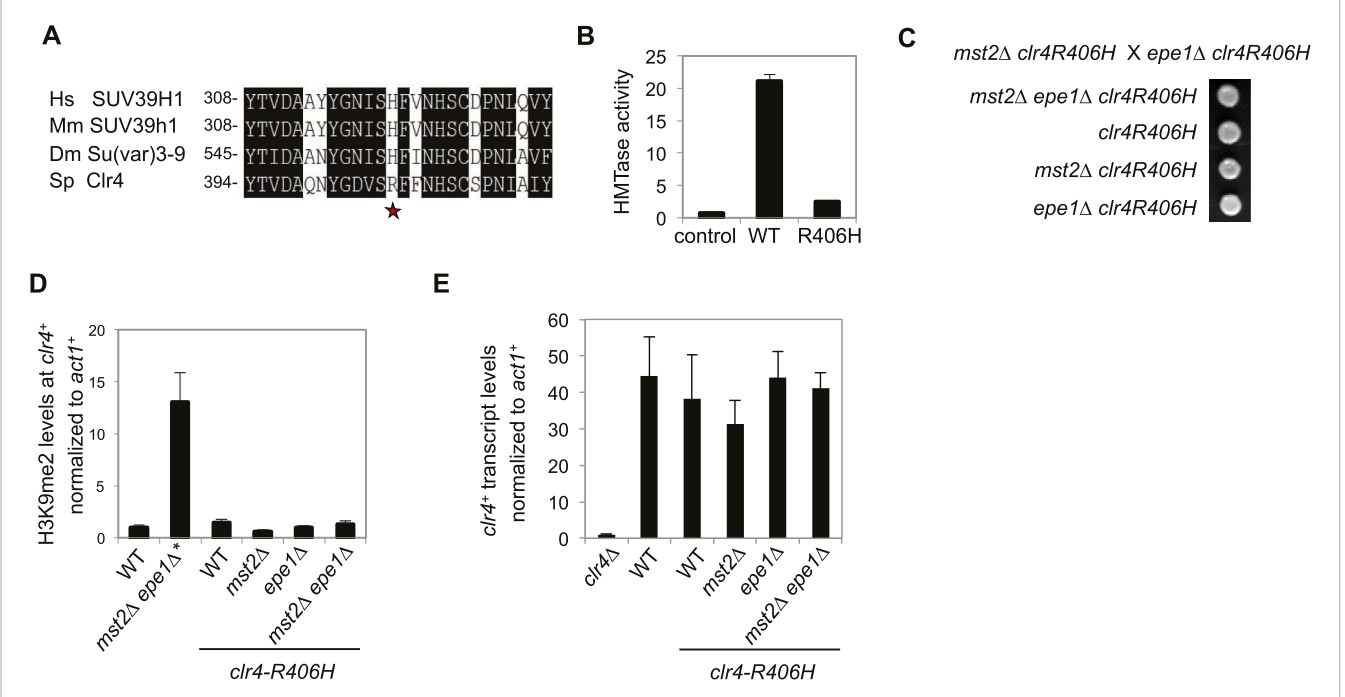

**Figure 6**. The high activity of Clr4 leads to growth defects of *mst2Δ epe1Δ* cells. (**A**) Sequence alignment of part of Clr4 homologues. * indicates R406 of Clr4. (**B**) In vitro histone methyltransferase assays were performed with recombinant GST-Clr4 or GST-Clr4-R406H together with a histone H3 (1–21) peptide. For the control reaction, no Clr4 was added. (**C**) Tetrad dissection analysis of the indicated genetic cross. (**D**) ChIP-qPCR analysis of H3K9me2 levels at the *clr4+* coding region, normalized against *act1+*. Error bars represent standard deviation of three experiments. (**E**) qRT-PCR analysis of *clr4+* mRNA levels, normalized against *act1+*. Error bars represent standard deviation of three experiments.

The following figure supplement is available for figure 6:

**Figure supplement 1**. The effect of *clr4-R406H* on silencing of *otr::ura4+*.

Our results also revealed the functional redundancy of Mst2 and Epe1 in regulating heterochromatin spreading, which explains why heterochromatin spreading only occurs in a small population of cells and requires the overexpression of Swi6 to be efficiently detected (*Noma et al., 2006*; *Wang et al., 2013*). In the absence of both Mst2 and Epe1, heterochromatin spreading increases significantly, leading to the inactivation of essential genes and severe growth defects. Such strong survival pressure results in the selection of cells that can establish heterochromatin at the *clr4+* locus, leading to reduced transcription of *clr4+* and decreased Clr4 protein levels, thus allowing cells to reach a new equilibrium where heterochromatin assembly at regular locations is intact while the negative effects of heterochromatin spreading are mitigated (*Figure 7*). Although we cannot test the epigenetic profiles of individual cells, it is possible that *mst2Δ epe1Δ* cells initially generate varied epigenetic profiles, and only cells containing H3K9me at the *meu6-clr4* locus are clonally selected due to its beneficial effects on cell growth. The quick generation of this epigenetic suppressor also benefited from the stabilization of ectopic heterochromatin domains in *mst2Δ epe1Δ* cells. Once established, such ectopic heterochromatin can be inherited, but can also be quickly erased to allow cells to adapt to new conditions. Interestingly, when H3K9me at the *clr4+* locus is blocked, the survival pressure instead selects cells that can establish heterochromatin at the *rik1+* locus to similarly reduce heterochromatin-forming abilities. The flexibility in heterochromatin assembly allows cells and their subsequent generations to efficiently cope with changes in heterochromatin levels.

The formation of heterochromatin at the *clr4+* locus does not employ any of the known heterochromatin assembly pathways. The intact 3'-UTR region of *clr4+* seems to initiate heterochromatin through a novel mechanism, although such a mechanism must be very inefficient in wild-type cells given that no H3K9me is observed. It is interesting to note that the 3'-UTR of *clr4+*

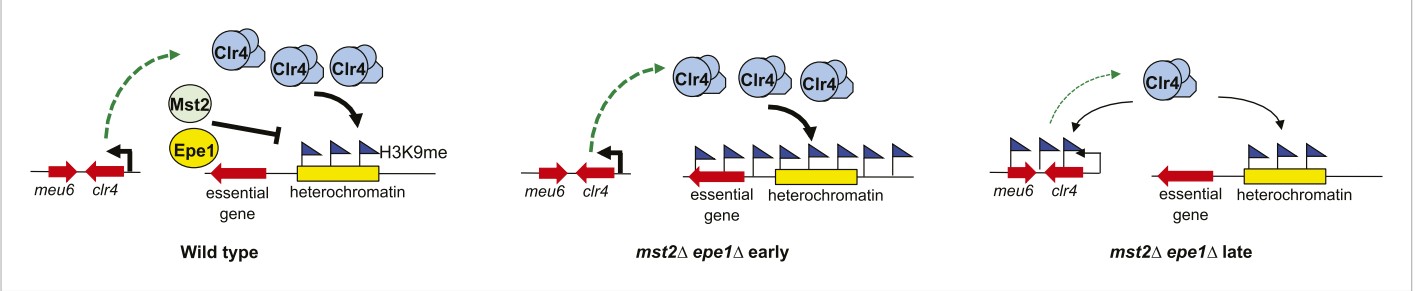

**Figure 7**. A model of the negative feedback of heterochromatin assembly.

overlaps the entire *meu6*[+] gene, which is arranged in a convergent orientation. The expression of *meu6*[+] is extremely low during vegetative growth, but is upregulated during meiosis, which might interfere with Clr4 expression, resulting in the loss of heterochromatin islands as seen in cells under conditions that induce meiosis (*Zofall et al., 2012*).

It has long been known that heterochromatin assembly in fission yeast is tightly regulated to prevent promiscuous heterochromatin assembly. For example, the amount of heterochromatin proteins such as Swi6 is limiting, and ectopic heterochromatin assembly can only succeed when Swi6 is overexpressed or when endogenous heterochromatin structures are compromised to release silencing proteins (*Iida et al., 2008*; *Kagansky et al., 2009*; *Tadeo et al., 2013*). Moreover, sub-telomeric heterochromatin regions, which contain no boundaries and no essential genes, can serve as 'sinks' to absorb extra heterochromatin proteins (*Tadeo et al., 2013*). These extensive pathways that limit heterochromatin assembly might be a response to the high enzymatic activity of Clr4, which is likely required for heterochromatin assembly in an otherwise highly compact and active genome. Our results provide an additional layer of control for cells to monitor heterochromatin levels through a negative feedback mechanism that uses the potentially promiscuous nature of heterochromatin assembly to induce heterochromatin formation at genes encoding heterochromatin assembly factors, thus ensuring the epigenetic stability of the genome.

Our results fit into a growing body of evidence demonstrating that epigenetic regulation of gene expression enables cells to adopt a wide variety of phenotypes to adapt to external or internal stresses (*Heard and Martienssen, 2014*). Compared to genetic mutations, epigenetic mutations provide much faster responses. Most importantly, the effects are reversible, allowing easy reversion to normal epigenetic profiles when external stimuli disappear. In cancer cells, such epigenetic variations might result in the inactivation of tumor suppressor genes during tumorigenesis and might also enable tumor cells to survive certain therapies (*Sharma et al., 2010*; *Kreso et al., 2013*). Therefore, our work sheds light on the mechanisms underlying how a relatively stable heterochromatic profile is maintained both under normal conditions and upon heterochromatin stress and will guide future efforts to combat epigenetic adaptations that interfere with cancer treatment.

## Materials and methods

### Fission yeast strains and genetic analyses

Detailed genotypes of strains used are listed in *Supplementary file 3*. Strains containing *meu6Δ* or *meu6-Flag* and *urg1-hht2-Flag* were constructed by a PCR-based module method. Genetic crosses were used to construct all other strains. For serial dilution plating assays, ten-fold dilutions of mid-log-phase cultures were plated on the indicated medium and grown for 3 days at 30°C.

### Genetic screen of the fission yeast deletion library

A two-step cross scheme was employed to avoid the accumulation of suppressors before colony growth measurement. Query strains (*mst2Δ::natMX6* and *epe1Δ::hphMX6*) were first separately mated with the fission yeast deletion library arrayed in 384 strains/plate format with the aid of the Singer RoToR HDA pinning robot as previously described (*Tadeo et al., 2013*). After mating and

selection, the resulting haploid double mutant libraries containing individual gene deletions with either mst2Δ or epe1Δ were mated again, and haploid triple mutants were selected on YES medium supplemented with antibiotics to measure cell growth.

## Chromatin immunoprecipitation (ChIP) analysis

ChIP analyses with H3K9me2 antibody (Abcam, Cambridge, MA) were performed as described previously (*Tadeo et al., 2013*). Quantitative real-time PCR (qPCR) was performed with Maxima SYBR Green qPCR Master Mix (Fermentas, Grand Island, NY) in a StepOne Plus Real-Time PCR System (Applied Biosystems, Grand Island, NY). DNA serial dilutions were used as templates to generate a standard curve of amplification for each pair of primers, and the relative concentration of the target sequence was calculated accordingly. An *act1* fragment was used as reference to calculate the enrichment of ChIP over WCE for each target sequence. Oligos used are listed in *Supplementary file 4*.

For histone turnover assay, cells were cultured in EMM–uracil medium, and then arrested for 4 hr by 20 mm HU, followed by the addition of 0.25 mg/ml uracil to induce the expression of H3-Flag, before ChIP analysis was performed.

ChIP–chip analyses were performed according to the 'Agilent Yeast ChIP-on-chip Analysis' protocol. The microarray used was an Agilent *S. pombe* Whole Genome ChIP-on-chip Microarray with additional probes that encompass centromeres, which were originally absent from the array due to the repetitive nature of these DNA sequences. At least two repeats were performed for each microarray experiment. To control for the experimental variation, the average of top 20 probes was set to 1 before averaging the results. For heterochromatin islands, the cutoff of H3K9me2 levels is 0.2. Microarray data have been deposited in the GEO database under accession number GSE60521.

## RNA analyses

Total cellular RNA was isolated from log-phase cells using MasterPure yeast RNA purification kit (Epicentre, Madison, WI) according to the manufacturer's protocol. Quantification with qRT-PCR was performed with Power SYBR Green RNA-to-CT one-step Kit (Applied Biosystems). RNA serial dilutions were used as templates to generate the standard curve of amplification for each pair of primers, and the relative concentration of target sequence was calculated accordingly. An *act1* fragment served as reference to normalize the concentration of samples. The concentration of each target gene in wild type was arbitrarily set to 1 and served as references for other samples. Oligos used are listed in *Supplementary file 4*.

## Acknowledgements

We thank Xavier Tadeo for help with SGA screen, Anudari Letian for technical assistance, James Manley, Hengbin Wang, and members of the Jia lab for helpful discussions and critical reading of the manuscript. This work was supported by National Institutes of Health grants R01-GM085145 to SJ. BDR was supported by NIH training grant T32-GM008798.

## Additional information

### Funding

| Funder | Grant reference | Author |
|---|---|---|
| National Institutes of Health (NIH) | R01-GM085145 | Songtao Jia |
| National Institutes of Health (NIH) | T32-GM008798 | Bharat D Reddy |

The funder had no role in study design, data collection and interpretation, or the decision to submit the work for publication.

### Author contributions

JW, BDR, Conception and design, Acquisition of data, Analysis and interpretation of data; SJ, Conception and design, Acquisition of data, Analysis and interpretation of data, Drafting or revising the article

## Additional files

### Supplementary files

• Supplementary file 1. List of heterochromatin peaks in different genetic background.

• Supplementary file 2. List of mutations identified in SGA screen.

• Supplementary file 3. Yeast strains used in this study.

• Supplementary file 4. Primers used in this study.

### Major dataset

The following dataset was generated:

| Author (s) | Year | Dataset title | Dataset ID and/or URL | Database, license, and accessibility information |
|---|---|---|---|---|
| Wang J | 2014 | Epigenetic adaptation to uncontrolled heterochromatin spreading | http://www.ncbi.nlm.nih.gov/geo/query/acc.cgi?acc=GSE60521 | Publicly available at NCBI Gene Expression Omnibus (GSE60521). |

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
