## [Decision Letter]

Thank you for submitting your manuscript titled “Rapid Epigenetic Adaptation to Uncontrolled Heterochromatin Spreading” for consideration for publication at *eLife*. Your manuscript was reviewed by three experts in the field and by a member of the Board of Reviewing Editors (BRE). The detailed reviews for your manuscript are included at the end of this letter. After a full discussion of the study and the reviews, we are happy to report that the reviewers and the BRE member found the study of interest to the journal and therefore we are happy to consider a revised manuscript. As you will see from the reviewers' comments, there are several minor points to be considered and need to be addressed. Also, the third reviewer has suggested 5 points under his/her major concerns. Upon discussion, we do not require you to address these points for publication, however, if you have data that can clarify the reviewers concerns, we suggest including such new data in the revised manuscript.

*Reviewer #1*:

In this study Wang et al. describe a role for the conserved Mst2 histone acetyltransferase in regulation of heterochromatin spreading and inhibition of ectopic heterochromatin in fission yeast. In addition, they discover that the combined loss of Mst2 and Epe1 results in uncontrolled heterochromatin spreading and the formation of ectopic heterochromatin leading to inactivation of essential genes. Most importantly, their results reveal an epigenetic adaptation mechanism that can lead to the inactivation of heterochromatic genes such as *clr4* and *rik1* in *mst2 epe1* double mutant cells that allows these cells to survive.

This is an exciting and important study that provides insight into the role of anti-silencing modifications in control of heterochromatin spreading. The results on adaptation of cells to loss of Mst2 and Epe1 are particularly interesting and reveal epigenetic, rather than genetic, inactivation as a major survival strategy in response to inappropriate heterochromatin spreading. This may be particularly relevant to cancer and other diseases linked to changes in chromatin modifications. The results raise many interesting questions that will stimulate new lines of study and will be of great interest to the field. The main conclusions of the paper are supported by a series of straightforward and beautifully executed experiments. I am happy to support publication in *eLife*.

Minor comment:

In the subsection headed “Mst2 and Epe1 are required to counteract the high activity of Clr4”, the authors state: “these cells are able to efficiently silence an *otr::ura4*^+^ reporter gene.” No data is shown to support this claim and should probably be added to the supplemental section.

*Reviewer #2*:

Wang et al. described a new role of the fission yeast Mst2 protein, a subunit of a HAT complex in the regulation of histone turnover at the heterochromatin regions, heterochromatin spreading and ectopic heterochromatin assembly. Deletion of mst2+ resulted not only in increased of heterochromatin spreading and silencing near telomeres but also increased ectopic heterochromatin assembly. Interestingly, the loss of Mst2 has similar phenotype in heterochromatin assembly as the loss of the anti-silencing factor Epe1. Further analysis revealed that the combined loss of Mst2 and Epe1 anti-silencing factors leads to widespread H3K9mehtyulation and growth defects due to gene inactivation. Surprisingly, the *mst2Δ epe1Δ* cells recover by accumulation of heterochromatin at genes required for heterochromatin assembly particularly clr4^+^.

My main concern is that the suppression of the *mst2∆ epe1∆* phenotype via silencing of clr4^+^ is a phenomenon that, while interesting and cool, is entirely specific to a mutant situation. Presumably, this never occurs in wild-type cells. In that sense, this paper is more about investigating a mutant pathology than revealing how wild-type cells function.

Minor points:

1) The authors state that the constitutive domains at the centromeres in mst2Δ are in good agreement with the wild type. This statement is only true for the chromosome 1. According to the Figure 1, we can clearly observe a decrease of the H3K9me signal at the centromere 2 and 3 in *mst2Δ* compare to the wild type. How can the authors explain these discrepancies?

2) The authors' *epe1Δ* is at odds with those of Zofall (58) who reported subtelomeric spread in *epe1Δ* cell.

3) How do the authors explain that only few heterochromatics islands can show a spread of H3K9me in the triple mutants *mst2Δ epe1Δ swi6Δ*?

4) Is RNAi required for ectopic silencing establishment?

5) In the Introduction, the authors should indicate that the establishment of constitutive heterochromatin at repetitive DNA elements requires RNA interference at the centromeres but not at other element such as telomeres (37).

6) The authors do not mention the recruitment of RNAi to these specific meiotic genes by the Mmi RNA surveillance machinery (17).

7) The authors state that they identified a few heterochromatic islands with a low level of H3K9me. This analysis should be explained more clearly in the Methods.

8) For a better understanding, the authors should add on the x axis the title “Chromosome position” on the ChIP-chip figures.

9) In the subsection headed “Misregulation of heterochromatin affects the fitness of *mst2∆ epe1∆* cells”, the two mutants *mst2Δ* and *epe1Δ* appear to have a very different phenotype on telomeres (Figure 1).

10) Figure 2, how many days of colony growth?

11) Figure 2, the time of growth is not described.

12) Figure 2, how did the RNAi factors score in this screen?

13) At the end of the subsection headed “Increased heterochromatin spreading is responsible for the initial growth defects of *mst2∆ epe1∆* cells”, add *clr4*^*+*^ locus.

14) Figure 4, *mst2Δ Flag-clr4*^*+*^ is missing.

*Reviewer #3*:

This manuscript describes the unrestrained assembly of heterochromatin via H3K9 methylation in fission yeast devoid of the H3K14 acetyltransferase Mst2 and the putative histone/protein demethylase Epe1. Epe1 may be an H3K9 demethylase and H3K14 acetylation is linked with transcription and histone turnover. Both activities act in parallel pathways to limit the extent of heterochromatin domains.

Interestingly, *mst2 epe1* double null cells are initially very slow growing but suppressors rapidly arise allowing normal growth. They show that natural suppressors result in silencing of the gene encoding the Clr4 H3K9 methyltransferase due to the formation of heterochromatin over the *clr4* gene itself. The *rik1* gene (a component of Clr4 methyltransferase complex) is similarly silenced in *mst2 epe1* mutant cells when clr4 cannot be silenced. Although the mechanism of silencing of the clr4 gene (and *rik1* gene) is not known, alteration of the region flanking the 3' end of the *clr4* gene inhibits heterochromatin assembly. Overall the results are interesting, however more specific information regarding how *meu6-clr4* locus is actually silenced would strengthen the manuscript.

Issues to be addressed:

1) The histone turnover experiment (Figure 1) would benefit from the inclusion of total H3 control ChIP in WT and mutant backgrounds. Also, since histone turnover is known to be much greater on highly transcribed genes such as *act1*, it would be wise to also compare turnover levels at *dh* to repressed genes (e.g. *nmt1* or other) where histone turnover should be low and thus provide a more accurate comparison. This would also provide an internal control to confirm that differences in histone turnover levels at different loci are being detected (i.e. *act1* v *nmt1*).

2) What is the frequency of fast growing colonies with *clr4*^*+*^ epialleles (and *rik1*^*+*^ epialleles) as opposed to the frequency of genuine mutations that inactivate clr4 or genes encoding other heterochromatin components?

3) Quantitative PCR ChIP, RT-PCR, ChIP-chip: It is not immediately clear from figures how ChIP signals were quantified and what is presented. My assumption was that it is %IP shown. Only when I read the methods/legends did I discover that all values are normalized relative to the *act1* gene. For ChIP the Y axis in all figures should be clearly labelled H3-FLAG/H3K9me2/ at *dg/clr4/rik1* relative to *act1* (or H3K9me2 ratio *dg/act1* etc.). Likewise, *clr4/rik1* transcript/mRNA relative to act1 in all relevant figures.

For ChIP-chip analyses, details should be provided for how H3K9me2 profiles were obtained and the methods utilized should conform with normally accepted practices for microarray data processing and presentation. From Figures 1, 2 and 3, it looks as if all values may be plotted relative to a control locus? The processing employed and units used need to be clearly described and indicated.

4) The data in Figure 5 is discussed in a section entitled: “The mechanism of heterochromatin assembly at the clr4^+^ locus”. Here the authors just show that manipulation of sequences flanking the 3' end of the *clr4* gene prevents heterochromatin assembly over *clr4* in *mst1 epe1* double null cells. They find that *dcr1*/RNAi, Mmi1/exosome and Pab2/3' end processing pathways are not involved. Red1 should be also tested as it is more directly involved in the recruitment of Clr4 to some heterochromatin islands. It would also make sense to test Rrp6 and Mtl1.

No explanation is given for how they conclude from RNA-seq data (Figure 5) that the *clr4* transcript runs through the convergent *meu6* gene. Is the same sized transcript detected by northern with *meu6* and *clr4* probes? Is 5' RACE consistent with a *clr4* (sense)-*meu6* (antisense) RNA being synthesized? Regardless, the experiments presented do not actually reveal the mechanism of heterochromatin assembly at the *clr4* locus in *mst2 epe1* null cells. How does this 3' region nucleate heterochromatin? If it is something to do with the 3' UTR of the *clr4* RNA then a more precise insertion of a known transcription terminator downstream of the Clr4 ORF should prevent heterochromatin assembly.

Does the *clr4* locus have anything in common with the rik1 locus and other H3K9me2 islands detected in *mst1 epe1 swi6* null cells? Apart from *meu6-clr4*, are there other H3K9me2 islands specific to *mst1 epe1 swi6* null cells that might be informative with respect to mechanism?

Related to this, did the screen of the deletion library (Figure 2) identify any additional genes that suppress the *mst2 epe1* slow growth phenotype but do not correspond to known heterochromatin components? If so, presumably they might suggest how *meu6-clr4* is silenced and these should at least be discussed.

5) More extensive analysis is required to allow the authors to conclude that *clr4-R406H* “mildly affected silencing” and that “these cells are able to efficiently silence the *otr::ura4*^*+*^ reporter”. A simple 5-FOA plate, with no -ura4 plate for comparison, is not sufficient. Are H3K9me2 levels of *dg/dh* repeats reduced? Are *dg/dh* transcript levels increased?

---

## [Author Response]

Reviewer #1:

*In the subsection headed “Mst2 and Epe1 are required to counteract the high activity of Clr4”, the authors state:* “*these cells are able to efficiently silence an* otr::ura4^+^
*reporter gene.*” *No data is shown to support this claim and should probably be added to the supplemental section*.

We have removed this claim.

Reviewer #2:

*My main concern is that the suppression of the* mst2 ∆epe1∆ *phenotype via silencing of* clr4^+^
*is a phenomenon that, while interesting and cool, is entirely specific to a mutant situation. Presumably, this never occurs in wild-type cells. In that sense, this paper is more about investigating a mutant pathology than revealing how wild-type cells function*.

Although the generation of an epigenetically silenced *clr4*^*+*^ allele only happens in *mst2∆ epe1∆* background, we can extrapolate that any other mutations or environmental insults resulting in a massive increase in heterochromatin assembly can trigger a similar response. Moreover, our results demonstrate the functional redundancy of Mst2 and Epe1 in controlling heterochromatin assembly in wild type cells.

*Minor points*:

*1) The authors state that the constitutive domains at the centromeres in mst2Δ are in good agreement with the wild type. This statement is only true for the chromosome 1. According to the*
Figure 1*, we can clearly observe a decrease of the H3K9me signal at the centromere 2 and 3 in* mst2Δ *compare to the wild type. How can the authors explain these discrepancies*?

We have included detailed microarray data around centromere 2 and 3 in Figure 1—figure supplement 1. The apparent higher amount of H3K9me2 at these regions in wild type cells was due to higher values of only a few probes. For the majority of probes, the values are very similar between wild type and *mst2∆* cells. We also used thinner lines in Figure 1 to better visualize this effect.

*2) The authors'* epe1Δ *is at odds with those of Zofall (*[58]*) who reported subtelomeric spread in* epe1Δ *cell*.

We have included detailed microarray data of the four telomeric regions of chromosome I and II (Figure 1—figure supplement 1). In all cases, the effect of *epe1∆* on telomeric heterochromatin is very minor. We currently do not know the reason for the discrepancy with [58]. It is known that *epe1∆* shows variable phenotypes in centromere silencing (48), suggesting that there might be two different sub-populations of cells with different behaviours. We have added a description of this discrepancy in the text (in the second paragraph of the subsection headed “Mst2 regulates histone turnover at heterochromatin”).

*3) How do the authors explain that only few heterochromatics islands can show a spread of H3K9me in the triple mutants* mst2Δ epe1Δ swi6Δ?

We observed spreading of H3K9me at the majority of heterochromatic islands in *mst2∆ epe1∆ swi6∆* cells. Due to space limitations, we only showed one in the original version. We have now included detailed data of all heterochromatin islands in Figure 3—figure supplement 1.

*4) Is RNAi required for ectopic silencing establishment*?

We found that RNAi is not required for ectopic silencing establishment at the *clr4*^*+*^ locus. We crossed *mst2∆ dcr1∆* with *epe1∆ dcr1∆* and found that all *mst2∆ epe1∆ dcr1∆* cells can still recover, consistent with our ChIP analysis that there is H3K9me at the *clr4*^*+*^ locus in these cells (Figure 5—figure supplement 1).

*5) In the Introduction, the authors should indicate that the establishment of constitutive heterochromatin at repetitive DNA elements requires RNA interference at the centromeres but not at other element such as telomeres (*[37]*)*.

We disagree with the reviewer's statement. It has been shown that the DNA repeats at telomeres contribute to heterochromatin establishment (24). The ability to establish heterochromatin at telomeres in RNAi mutant, as described in [37], is because of redundant heterochromatin establishment pathways at telomeres (24).

*6) The authors do not mention the recruitment of RNAi to these specific meiotic genes by the Mmi RNA surveillance machinery (*[17]*)*.

We added a sentence in the fourth paragraph of the Introduction to describe the recruitment of RNAi to meiotic genes.

*7) The authors state that they identified a few heterochromatic islands with a low level of H3K9me. This analysis should be explained more clearly in the Methods*.

We stated in the Methods section the specific cut off used to determine heterochromatin islands. A list of the heterochromatin islands was included in [Supplementary-material SD1-data].

*8) For a better understanding, the authors should add on the x axis the title* “*Chromosome position*” *on the ChIP-chip figures*.

We added axis label “Chromosome position” for all ChIP-chip figures.

*9) In the subsection headed “Misregulation of heterochromatin affects the fitness of* mst2∆ epe1∆ *cells”, the two mutants* mst2Δ *and* epe1Δ *appear to have a very different phenotype on telomeres (*Figure 1*)*.

Even though *mst2∆* and *epe1∆* have similar effects on histone turnover, ectopic heterochromatin assembly (Figure 1), and on suppression of RNAi mutants ([39] and [48]), they behaved differently in the spreading of heterochromatin domains at telomeres and centromeres (Figure 1), as well as heterochromatin maintenance in the absence of initiation signals (37), suggesting that these two mutants are not identical.

*10)*
Figure 2*, how many days of colony growth*?

The picture shown in Figure 2 is 6 days of growth after dissection of tetrads. We added the description in the figure legend.

*11)*
Figure 2*, the time of growth is not described*.

The dilution analyses were performed after 1 day of growth in rich medium. We added the description in the figure legend.

*12)*
Figure 2*, how did the RNAi factors score in this screen*?

RNAi factors were not identified in our screen. We added a description of this fact in the text (in the subsection headed “Misregulation of heterochromatin affects the fitness of *mst2∆ epe1∆* cells”). We also showed a tetrad dissection of an *mst2∆ dcr1∆* and *epe1∆ dcr1∆* cross in Figure 2—figure supplement 2.

*13) At the end of the subsection headed “Increased heterochromatin spreading is responsible for the initial growth defects of* mst2∆ epe1∆ *cells”, add* clr4^+^
*locus*.

We changed the text to “*mei4*^*+*^ and *clr4*^*+*^ loci”.

*14)*
Figure 4*,* mst2Δ Flag-clr4^+^
*is missing.*

We added a panel in Figure 4 to show different combinations. There are no growth defects associated with *mst2∆ Flag-clr4*.

Reviewer #3:

*1) The histone turnover experiment (*Figure 1*) would benefit from the inclusion of total H3 control ChIP in WT and mutant backgrounds. Also, since histone turnover is known to be much greater on highly transcribed genes such as* act1*, it would be wise to also compare turnover levels at* dh *to repressed genes (e.g.* nmt1 *or other) where histone turnover should be low and thus provide a more accurate comparison. This would also provide an internal control to confirm that differences in histone turnover levels at different loci are being detected (i.e.* act1 *v* nmt1*)*.

We normalized our data to a non-transcribed mating-type region, which showed low histone turnover (3), in new Figure 1. The results are similar to those without such normalization.

*2) What is the frequency of fast growing colonies with* clr4^+^
*epialleles (and* rik1^+^
*epialleles) as opposed to the frequency of genuine mutations that inactivate* clr4 *or genes encoding other heterochromatin components*?

Our data suggest that the *clr4*^*+*^ epiallele is much more prevalent than genuine genetic mutations. For example, experiments in Figure 4 showed that the suppressor is epigenetic rather than genetic, as cells inherited the silenced *clr4*^*+*^ epiallele can revert back to normal. The results shown are representative of multiple independent suppressor strains. The accurate determination of the frequency of epiallele vs. genuine mutations requires genome sequencing of independent suppressor strains.

*3) Quantitative PCR ChIP, RT-PCR, ChIP-chip: It is not immediately clear from figures how ChIP signals were quantified and what is presented. My assumption was that it is %IP shown. Only when I read the methods/legends did I discover that all values are normalized relative to the* act1 *gene. For ChIP the Y axis in all figures should be clearly labelled H3-FLAG/H3K9me2/* at dg/clr4/rik1 *relative to* act1 *(or H3K9me2 ratio* dg/act1 *etc). Likewise,* clr4/rik1 *transcript/mRNA relative to* act1 *in all relevant figures*.

*For ChIP-chip analyses, details should be provided for how H3K9me2 profiles were obtained and the methods utilized should conform with normally accepted practices for microarray data processing and presentation. From*
Figures 1, 2 and 3*, it looks as if all values may be plotted relative to a control locus? The processing employed and units used need to be clearly described and indicated*.

We added labels in all relevant figures when a control locus was employed for normalization. For ChIP-chip analysis, we normalized data in order to average results from different experiments. We explained in the Methods section about the details of data processing.

*4) The data in*
Figure 5
*is discussed in a section entitled:* “*The mechanism of heterochromatin assembly at the* clr4^+^
*locus*”*. Here the authors just show that manipulation of sequences flanking the 3' end of the* clr4 *gene prevents heterochromatin assembly over* clr4 *in* mst1 epe1 *double null cells. They find that* dcr1*/RNAi, Mmi1/exosome and Pab2/3' end processing pathways are not involved. Red1 should be also tested as it is more directly involved in the recruitment of Clr4 to some heterochromatin islands. It would also make sense to test Rrp6 and Mtl1*.

We changed the section title to: “Sequence 3’ to *clr4*^*+*^ is required for heterochromatin assembly at the *clr4*^*+*^ locus in *mst2∆ epe1∆* cells”. We attempted to generate triple mutations of *mst2∆ epe1∆ red1∆* and *mst2∆ epe1∆ red1∆ rrp6∆.* However, we were unable to generate such strains. Both *red1∆* and *rrp6∆* cells are extremely sick, making it difficult to determine whether the lethality is due to the cumulative sickness of these strains or because such strains cannot generate clr4^+^ epialleles. *Mtl1* is an essential gene, and we currently do not have *mtl1* mutant.

*No explanation is given for how they conclude from RNA-seq data (*Figure 5*) that the* clr4 *transcript runs through the convergent* meu6 *gene. Is the same sized transcript detected by northern with* meu6 *and* clr4 *probes? Is 5' RACE consistent with a* clr4 *(sense)-*meu6 *(antisense) RNA being synthesized? Regardless, the experiments presented do not actually reveal the mechanism of heterochromatin assembly at the* clr4 *locus in* mst2 epe1 *null cells. How does this 3' region nucleate heterochromatin? If it is something to do with the 3' UTR of the* clr4 *RNA then a more precise insertion of a known transcription terminator downstream of the Clr4 ORF should prevent heterochromatin assembly*.

*Does the* clr4 *locus have anything in common with the rik1 locus and other H3K9me2 islands detected in* mst1 epe1 swi6 *null cells? Apart from* meu6-clr4*, are there other H3K9me2 islands specific to* mst1 epe1 swi6 *null cells that might be informative with respect to mechanism*?

Our RNA-seq data is strand specific, which clearly indicated that the *clr4*^*+*^ transcript covers the *meu6*^*+*^ coding sequence and that the *meu6*^*+*^ mRNA levels are very low. Such a conclusion is consistent with the annotation of genome in Pombase based on results of Marguerate et al., 2012 and Rhind et al., 2011. Therefore, we believe a 5'-RACE experiment is unnecessary.

We currently do not know the mechanism that induces heterochromatin assembly at the *clr4*^*+*^ locus, except that sequences 3' to *clr4*^*+*^ is required for heterochromatin assembly. But we did rule out most of the known heterochromatin establishment mechanisms, such as RNAi, Mmi1, and convergent transcription, indicating that a novel mechanism is involved. We are actively dissecting the mechanisms that establish this ectopic heterochromatin, including precisely manipulating transcription termination at the *clr4*^*+*^ locus, and we believe such an experiment is out of the scope of the current study.

We found that a common theme of the *clr4*^*+*^ and *rik1*^*+*^ loci is that both are adjacent to genes up-regulated during meiosis, although whether there is any causal relationship is still unknown. Apart from *meu6-clr4*, there are other H3K9me2 islands in *mst2∆ epe1∆ swi6∆* cells, such as *mug8*, *SPAC8c9.04*, and *sol1*. However, *meu6-clr4* does not show accumulation of H3K9me in *mst2∆ epe1∆ swi6∆* cells. Therefore, they are not likely to be formed through similar mechanisms.

*Related to this, did the screen of the deletion library (*Figure 2*) identify any additional genes that suppress the* mst2 epe1 *slow growth phenotype but do not correspond to known heterochromatin components? If so, presumably they might suggest how* meu6-clr4 *is silenced and these should at least be discussed*.

There are a number of additional mutants identified, which are listed in [Supplementary-material SD2-data]. Due to the fast generation of the *clr4*^*+*^ epiallele, such an experiment is intrinsically noisy and many of the identified factors are simply noise. The screen is not designed to identify factors required for heterochromatin assembly at the *clr4*^*+*^ locus. Theoretically, such a mutation will result in cells that cannot recover, which can be tested by repeated pinning of the cells to measure their growth. However, there are many strains in the library causing growth defects, and the linkage of genes to three different loci will result in many SGA spots showing growth defects, thus the noise will be too high for such experiments to be informative.

*5) More extensive analysis is required to allow the authors to conclude that* clr4-R406H “*mildly affected silencing*” *and that* “*these cells are able to efficiently silence the* otr::ura4^+^
*reporter*”*. A simple 5-FOA plate, with no -ura4 plate for comparison, is not sufficient. Are H3K9me2 levels of* dg/dh *repeats reduced? Are* dg/dh *transcript levels increased*?

We have removed this claim. We have included RT-PCR analysis of *otr::ura4*^*+*^ transcript levels, shown in Figure 6—figure supplement 1.